# Early Nutrition eAcademy Southeast Asia e-Learning for Enhancing Knowledge on Nutrition during the First 1000 Days of Life

**DOI:** 10.3390/nu12061817

**Published:** 2020-06-18

**Authors:** Brigitte Brands, Sirinuch Chomtho, Umaporn Suthutvoravut, Christopher Chiong Meng Boey, Swee Fong Tang, Keith M. Godfrey, Berthold Koletzko

**Affiliations:** 1Dr. von Hauner Children’s Hospital, University Hospital, LMU—Ludwig-Maximilians-Universität München, D-80377 Munich, Germany; Berthold.Koletzko@med.uni-muenchen.de; 2Pediatric Nutrition STAR, Department of Pediatrics, Faculty of Medicine, Chulalongkorn University, Bangkok 10330, Thailand; schomtho@gmail.com; 3Department of Pediatrics, Faculty of Medicine Ramathibodi Hospital, Mahidol University, Bangkok 10400, Thailand; u.suthut@gmail.com; 4Faculty of Medicine, University of Malaya, Kuala Lumpur 50603, Malaysia; boeycm@ummc.edu.my; 5Department of Paediatrics, Faculty of Medicine, Universiti Kebangsaan Malaysia, Bangi 43600, Malaysia; sweefongtang@gmail.com; 6MRC Lifecourse Epidemiology Unit and NIHR Southampton Biomedical Research Centre, University of Southampton, University Hospital Southampton NHS Foundation Trust, Southampton SO16 6YD, UK; kmg@mrc.soton.ac.uk

**Keywords:** first 1000 days, early nutrition, malnutrition, lifelong health, e-learning, healthcare professionals, CME, CPD, southeast Asia

## Abstract

Background: The double burden of both under- and overnutrition during the first 1000 days is highly prevalent in Southeast Asia (SEA), with major implications for lifelong health. Tackling this burden requires healthcare professionals (HCPs) to acquire evidence-based current knowledge and counselling skills. We assessed the needs of HCPs in SEA and developed a continuing medical education/professional development (CME/CPD) program using an e-learning platform to reduce existing gaps. Methods: European, Thai and Malaysian universities collaborated with SEA national nutrition associations in the Early Nutrition eAcademy Southeast Asia (ENeA SEA) project. We assessed HCPs’ needs using questionnaires and mapped CME/CPD programmes and regulations through stakeholder questionnaires. Using a co-creation approach, we established an e-learning platform. Evaluation in users was undertaken using questionnaires. Results: HCPs in SEA reported major training gaps relating to the first 1000 days of nutrition and limited impact of existing face-to-face training. Existing pre/postgraduate, residency and CME/CPD programmes did not adequately address the topic. To address these gaps, we produced a targeted e-learning platform with six modules and CME-tests. National ministries, Thai and Malaysian universities, and professional associations endorsed the training platform. To date, over 2600 HCPs have registered. Evaluation shows high acceptance and a very positive assessment. Conclusions: Dedicated e-learning can reduce major gaps in HCP training in SEA regarding nutrition during the first 1000 days of life at scale and is highly valued by both users and key stakeholders.

## 1. Introduction

Despite encouraging economic development, considerable nutritional challenges remain across Southeast Asia (SEA). In children and adults, obesity prevalence has rapidly increased, while underweight in young children, high rates of stunting and of micronutrient deficiencies continue to exist, resulting in a double or even triple burden of malnutrition [1,2]. The burden of diabetes and other noncommunicable diseases (NCDs) is increasing rapidly and is now a major public health concern, with substantial adverse consequences for health, wellbeing, productivity, and economic development [3]. Thailand and Malaysia are increasingly urbanized, with rural population decrease rates expected from 2018 until 2050 of −42,3 % and −31,1%,respectively, associated with a shift from infectious disease to a “western” chronic disease burden [4,5].

Early life nutrition during preconception, pregnancy, infancy and early childhood is a significant contributor to the lifelong NCD burden, through a “programming” effect on the later risk of NCD. Research in the field of early nutrition and lifestyle during the first 1000 days of life after conception demonstrated that optimised nutrition and lifestyle during this critical phase of early development provides a window of opportunity to promote life-long health and wellbeing [6,7].

An important measure to tackle the double burden of malnutrition is providing health care professionals (HCPs) with current, evidence-based knowledge on early nutrition and lifestyle recommendations that enables effective counselling of (pre-) pregnant women and of families with infants and young children. Nutrition-related health problems including low birth weight, protein-energy malnutrition, iron deficiency anaemia and common deficiencies of micronutrients such as iron, iodine and vitamins A and D can be addressed through targeted advice from HCPs with appropriate knowledge and skills. Every encounter a HCP has with their patient is a window of opportunity for positive change. The World Health Organisation (WHO) has set SEA region guidelines for continuing medical education/professional development (CME/CPD) [8], but lack of motivation and a paucity of needs–based accredited CME/CPD programmes and incentives for their use has limited the impact in Thailand and Malaysia. E-learning formats have the potential to markedly increase the reach of CME/CPD, but no such programme targeting early nutrition and lifestyle for SEA HCPs has been made available.

E-learning is a convenient and cost-efficient learning strategy, particularly in an era of social distancing in pandemic circumstances. It is easily accessible from various locations and devices and allows studying at the user’s own pace and time availability, with no time and cost investment for travel to a learning site. Therefore, we chose to establish e-learning on early nutrition and lifestyle for HCPs in SEA. Here we report on the approach used, the assessment of needs, and the experience with the established e-learning platform.

## 2. Materials and Methods

### 2.1. The Early Nutrition eAcademy Southeast Asia Project (ENeA SEA)

The ENeA SEA project started on 15 October, 2016, co-funded by the European Union (EU) Erasmus + Capacity-Building for Higher Education (CBHE) Programme with a contribution of nearly 1 Million EUR for 3.5 years. The ENeA SEA consortium consists of leading experts in the field of early nutrition and e-learning from nine institutions: three universities from Europe, four universities from Thailand and Malaysia, the Nutrition Association in Thailand and the Nutrition Society in Malaysia (Figure 1). The goal of the consortium is to address unmet needs for development and implementation of a scientific evidence-based, unbiased and open access early nutrition e-learning programme suitable for CME and CPD for practicing HCPs in SEA. The focus was placed on providing practically applicable knowledge and skills on early nutrition and lifestyle during the critical first 1000 days of human development.

Medical doctors that provide care for women who may get pregnant or are pregnant and for families with infants and young children were the primary target HCP group in the project, because they act as “multipliers” to set standards in the counselling process among HCPs at different levels. We anticipated that establishing knowledge among these professionals would create interest among other HCP groups, including nurses, midwives and nutritionists. Therefore, e-learning content was written at the level required for medical doctors, serving as the basis for subsequent adaptation for other HCP groups.

### 2.2. Assessment of HCPs’ Needs and CME/CPD Landscape in SEA

Analysing the situation and assessing the needs of HCPs in SEA, with a focus on Thailand and Malaysia, was a prerequisite for achieving the project outputs, including
Identifying specific topics and their prioritisation to develop a targeted ENeA SEA curriculum and e-learning modulesCreating appropriate, effective educational and technical formats optimally matching the needs of target groupsEnsuring that the content was of high practical relevance and could be directly translated into clinical practice and counselling, and into CME/CPD programmes, complementing existing training activities.


Understanding the needs of Thai and Malaysian HCPs related to early nutrition and lifestyle and analysing the CME/CPD structures were achieved through desktop research combined with online surveys containing closed and open questionnaire elements. The focus was to identify the underlying gaps in the transfer of knowledge in doctor’s practice to (pre-) pregnant and lactating women, infants and children (e.g., lack of counselling competencies), and the reasons for these gaps. Alongside identifying gaps in knowledge in relation to early nutrition and lifestyle healthcare counselling, the questionnaires sought to examine outdated knowledge and practices.

### 2.3. Desktop Research

Desktop research was performed to study and understand the present landscape of CME/CPD systems in Thailand and Malaysia, and more broadly in the SEA region. Using the Google search engine, the terminologies searched comprised: CME/CPD in Southeast Asia, Medical Association of Thailand, Medical Council of Thailand, Medical Association of Malaysia, WHO Guidelines, ASEAN Recommendations. The documents and websites analysed are listed in Table A1.

### 2.4. Online Surveys

The content for questions of the online surveys was discussed among consortium partners, doctors in practice and nutritionists/dieticians working in the field of early nutrition, leaders of regional nutrition and medical societies. A survey containing three questionnaires was set up, discussed and approved by the EU and SEA partners. The questionnaires contained questions of different types, including dichotomous closed questions (Yes/No), Likert response scale (bipolar rating 1 to 5), nominal questions (multiple selection options) and open questions (text fields). All three questionnaires contained consent to terms and conditions of participation, collection of sociodemographic data, closed and/or open question, and agreement to publish anonymised survey results.

The questionnaires were administered to HCPs for assessment of their needs (bottom-up approach), leaders of professional associations and key stakeholders holding leadership positions at SEA universities and in governmental bodies (top-down approach). Closed questions were quantitatively analysed based on automatically generated reports provided by the survey software. Open questions were analysed thematically and manually by two researchers after the raw data was downloaded from the online survey.

To efficiently distribute, collect and analyse the responses to the questionnaires, the online tool “Surveymonkey” was chosen (https://www.surveymonkey.com/; SurveyMonkey Europe UC, Dublin, Ireland). The questionnaires were beta-tested by SEA project partners, resulting in further optimisation of the questions. The questionnaire links together with an accompanying introduction of the study’s context and aims, and detailed consent to terms and conditions of participation were provided to all SEA project partners who distributed the information to their personal and professional networks in spring 2017 to target HCPs in the field of early nutrition. Inclusion criteria for being eligible to fill-in the online questionnaires were set by the mandatory selection of a health care profession from a drop down menu. In addition, only HCPs from the SEA region were included in the analysis. There were no further inclusion restrictions to allow for a higher number of participation and to retrieve a broader spectrum of answers.

### 2.5. Development of ENeA SEA e-Learning Platform, Curriculum and e-Modules and its Evaluation

Data from the online surveys were analysed to determine teaching and learning contents to provide the required knowledge and skills, and formats that meet the needs of Thai and Malaysian doctors. The data also provided the basis for guiding exploitation and sustainability measures to maximise the impact of the project outputs.

The Early Nutrition eAcademy Global (ENeA Global) platform of LMU Munich served as the basis for generating adapted and needs driven targeted content of the ENeA SEA modules, along with an established e-learning programme on malnutrition of the University of Southampton. By comparative analysis of the needs assessment and existing content of the ENeA Global platform, topics for e-learning modules and relevant content adaptations for SEA were determined. Guided by a “scientific content panel” and through an iterative regular review process, six e-learning modules were identified. A “Distance Learning Expert Panel” guided platform design using open source learning management software Moodle based on the existing ENeA Global Moodle infrastructure. Instructional design elements comprised videos, animated characters representing SEA regional healthcare staff and clients, case-based scenarios and interactive elements using H5P, a free and open-source content collaboration framework based on JavaScript. H5P is an abbreviation for HTML5 Package, and enables for everyone to create, share and reuse interactive HTML5 content. Innovation of the e-learning platform and its modules is reflected by an interactive and culturally sensitive design approach. It is based on constructivism, supporting the learner to construct knowledge based on their existing knowledge through situated learning and authentic tasks. A narrative counselling situation of a client and a HCP shown through a 3D video is such an example. Scenario-based cases and culturally appealing visual presentation motivate and actively engage the learner in their learning process, which is very important for effective learning. Furthermore, to achieve a high level of identification of the learner, great care was taken to design characters representative of SEA and to reflect real life situations. This facilitates the learner to build concrete knowledge through experiencing relevant case scenarios, which in turn makes it possible for the learner to apply their gained knowledge in practice.

### 2.6. Interdisciplinary, International Process of co-Creation and Capacity Building

Conceptualising, generating and implementing the e-learning platform with targeted curriculum and e-modules required a carefully steered process of co-creation, with close collaboration of experts from different disciplines and cultural backgrounds. Supported by a project steering committee, two panels comprising experts in the field of early nutrition and lifestyle and e-learning experts, respectively, guided the development and implementation of the e-learning platform, curriculum, contents and instructional design formats. Dedicated face-to-face working sessions and training weeks for co-creating contents and formats along with online meetings, feedback and review characterised the collaboration. Regular evaluation within the project consortium was undertaken to optimise collaboration.

In addition to a standard learning curriculum of modules with a structured learning pathway, a customised learning route was developed, enabling the user to generate their own curricula. This is based on actively selecting topics and automatic semantic analysis of an open text field where users can indicate their areas of interest. The software ReaderBench allows comparing and matching this information with all e-learning contents (78 lessons) offered on the platform in less than a few seconds, providing the user with individualised curriculum recommendations [9,10,11]. For both the standard and the customised learning routes, CME Certificates of Completion with European CME credits (EACCME accreditation) are offered as pdf files.

### 2.7. Descriptive and Explorative Evaluation of e-Learning Platform and Modules

A year after the launch of the e-learning platform and first modules, the descriptive user statistics (April 2020) and explorative evaluation questionnaires were analysed (January 2020) using Moodle inherent statistical and report generation plugins on the ENeA SEA e-learning platform. We obtained information on the number of registered users and country of origin, the number of users enrolled into each module, and the number of CME tests completed per module and their pass-rates. An explorative evaluation questionnaire to collect user feedback was generated and implemented at the end of each module as a mandatory element for the user to access the subsequent CME test and Certificate. The questionnaire contained 11 closed questions in addition to the collection of sociodemographic data (Table A2).

## 3. Results

### 3.1. Questionnaire 1: Online Survey with Closed Questions to HCPs

The first online questionnaire containing mainly closed questions was completed by 142 respondents. The respondent sample comprised pediatricians (*n* = 42), nutritionists (*n* = 44), dieticians (*n* = 14), other medical doctors (*n* = 12), scientists (*n* = 6), pharmacists (*n* = 3), nurses (*n* = 3) and other health care professionals (*n* = 18). Most participants (92%) were from Thailand and Malaysia. The most commonly requested topics for receiving CME were “nutrition and lifestyle during pregnancy”, “breastfeeding” and “nutritional care of preterm infants”. 85–91% of HCPs reported regular internet use. English language proficiency was higher among doctors than other HCP groups. 57% indicated a preference for e-learning over face-to-face training, preferably using a mobile device rather than a desktop computer. About 80% of HCPs were part of a CME CPD system, and 58% rated the importance of CME/CPD credits as high or very high, with national credits reported as more important than European or credits. The questionnaire and its full results are presented in Table 1.

### 3.2. Questionnaire 2: Online Survey with Open Questions to HCPs

The second survey with open text fields to questions was completed by 105 participants. Some 64% of participants were from Thailand and 28% from Malaysia; 90% practised in urban areas. Participants comprised pediatricians (*n* = 30), nutritionists (*n* = 30), other medical doctors (*n* = 13), dieticians (*n* = 11) and lecturers (*n* = 8). Themes of highest interest to users were “nutrition in pregnancy”, “infant nutrition and feeding”, “breastfeeding” and “best practice/guidelines/latest research findings on early nutrition and lifestyle”.

### 3.3. Questionnaire 3: Mixed Online Survey to Stakeholders in the CME/CPD Sector

The third questionnaire addressed to governing bodies and stakeholders influencing or determining CME/CPD was conducted in May 2017 and was completed by 11 participants; full results are presented in Table 2. Participants were from Thailand (*n* = 7) and Malaysia (*n* = 4), with eight participants holding responsibility for setting the standards for HCP professional specialisation, training or CME accreditation in the College of Obstetrics and Gynaecology, Academy of Medicine of Malaysia, the Medical Deans Council of the Malaysian Public Universities, the Malaysian College of Paediatrics, the Academy of Medicine of Malaysia, the Royal College of Pediatricians of Thailand, the Medical Council of Thailand, the Allergy, Asthma and Immunology Association of Thailand and the Royal College of Orthopaedic Surgeons of Thailand.

### 3.4. Review of the CME/CPD System in Thailand and Malaysia and in the SEA Region

There were wide variations in CME/CPD assessment in different SEA countries and healthcare systems. Centralised recertification examinations were often perceived as threatening and difficult to tailor to the needs of individual practitioners. Most systems are currently based on an hour-related credit system. There are common features of content and process that allow for international recognition of CME/CPD activities. Some countries in SEA have systems of mandatory periodic recertification. However, efforts to make CME/CPD programmes mandatory with an hours-based credit system for recertification have not yet been successful, mostly because of lack of a needs-based accredited CME/CPD programmes, incentives and a regulatory framework. Despite these limitations, different professional bodies, academic institutes and medical councils in many SEA countries have organised CME/CPD programmes and encourage doctors to actively participate in these. The SEA Regional Network of Medical Councils adopted a concept paper on “Continuing medical education” and developed Guidelines for CME for the SEA region in 2008 [8]. These guidelines specified minimum standards for CME/CPD activities for registered medical doctors. It invited national authorities to adapt the guidelines to their countries’ needs, with special emphasis on HCPs working in rural areas. However, designing good quality CME/CPD courses appropriate to the needs of different categories of practitioners was recognised as a substantial challenge requiring professional, technical and financial support. In Thailand, a formal CME/CPD qualification program has been re-established, and there is an on-going ambition to implement compulsory CME/CPD procedures.

### 3.5. Development of the ENeA SEA e-Learning Platform, Curriculum and Modules and Its Evaluation

The included modules address nutrition and lifestyle in pregnancy, breastfeeding, breastmilk substitutes, complementary feeding, nutritional care of preterm infants and malnutrition (Figure 2). The breast milk substitutes and nutrition and lifestyle in pregnancy modules have already been translated into the Thai language, and the remaining modules are currently being translated into both Thai and Malay to increase outreach and to overcome remaining language barriers. Associated partners in the ENeA SEA project supported the identification of key personnel and establishing the necessary contacts to work towards embedding the ENeA SEA curriculum and modules in existing pre- and postgraduate qualification programmes and into the local CME/CPD system. Moreover, the SEA partners contributed to establishing the requirements and processes for national accreditation. Feedback from SEA learners suggested that the interactive and culturally sensitive design approach works extremely well and achieves the intended learning effect. The evaluation results indicated that these interactive elements such as H5P were highly appreciated by the learners and contributed greatly to keeping the levels of learner engagement high, thereby enhancing the efficacy of the learning process.

#### Descriptive and Explorative Evaluation in Users

By April 2020, 2688 users had registered on the ENeA SEA Moodle platform, with the highest number enrolled in the breastfeeding module (*n* = 636), followed by the modules nutrition and lifestyle in pregnancy (*n* = 421), complementary feeding (*n* = 293), breastmilk substitutes (*n* = 266) and nutritional care of preterm infants (*n* = 216). The highest completion rate of a module was 72%. Comprehensive results are shown in Table 3.

Here we report on the results of the explorative evaluation questionnaires of the most popular module “breastfeeding”.

The evaluation questionnaire of this module was completed by a total of n = 245 users; amongst these, 76% were students/trainees, 17% Pediatricians, 3% General Practitioners and 3% Nutritionists/Dieticians. 98% of users agreed or strongly agreed that the module content was arranged in a clear and logical manner, 93% agreed or strongly agreed that the interface was easy to use. 96–99% of users agreed or strongly agreed that the module content was consistent with objectives, that this module met their needs in the field of early nutrition, that their confidence in counselling patients on breastfeeding has increased a result of this module and that this module will lead to changing their perspectives and/or practice. 97% of users would recommend this e-learning activity to others and agreed or strongly agreed that the content was relevant to Southeast Asia, while 92% would take another e-learning module based on this experience, 94% agreed or strongly agreed that the module has helped them to inform or train other HCPs. The module overall was rated by 54% of users with 9 or 10 out of 10 (1 being the minimum (poor) and 10 being the maximum score (excellent)) and by 38% of users with 7 to 8 out of 10. Some 7% of users rated the module overall with 5 or 6 out of 10.

The results of the evaluation questionnaires of the other three modules were completed by fewer participants (range from *n* = 53 up to 87), but all showed a similar pattern of responses to the “breastfeeding” module. While the collection of feedback from evaluation questionnaires is a valuable information source for further improving the ENeA SEA modules and programme, knowledge acquisition by the user is evaluated by the embedded CME tests with multiple-choice questions. Moreover, passing the CME tests with a threshold of 70% of correct answers is a pre-requisite to obtain the CME certificates of completion.

## 4. Discussion

While there are other e-learning programmes on the first 1000 days available globally, the ENeA SEA programme is the first e-learning qualification programme for HCPs focussing on nutrition and lifestyle during the first 1000 days of life especially for the SEA region. This ensures a targeted intervention meeting the special challenges and needs of HCPs in SEA. E-learning on the topics nutrition and lifestyle in pregnancy, breastfeeding and infant nutrition were of high interest to SEA HCPs. This is largely consistent with what decision-makers indicated as thematic priorities for inclusion into CME/CPD programmes in Thailand and Malaysia, i.e., complementary feeding, infant formula feeding, breastfeeding and nutrition of preterm infants. While English language proficiency was high among medical doctors, other HCP groups would benefit from local language versions of e-modules. Thus, translation of the platform and its modules into Thai and Malay languages has commenced and the first modules have been launched in the Thai language. Overall user feedback for the modules was very good. While the total number of registered users has increased up to 2688, the number of enrolled users in each module is not as high. This could be explained by the “on-demand” learning nature of the platform, which allows to start a module at any time after having registered on the platform. An increase is expected following dissemination and exploitation activities by SEA partners and the recently achieved endorsement and CME/CPD accreditation from the Thai Ministry of Health and Malaysian Medical Association and national professional associations. In addition to local language translation, a dedicated tutor area has been established on the platform to support teachers and trainers in the field with materials guiding them on how to embed and actively use ENeA SEA modules in their existing pre-and postgraduate programmes to further act as multiplicators increasing the outreach and update of ENeA SEA modules. This includes comprehensive Frequently Asked Questions (FAQ) and question banks, as well as concrete case studies which are accessible for tutors after registration and verification by the SEA lead partners.

However, the practical skills training, interactive case discussion and personal guidance element which would be needed for a comprehensive qualification programme cannot be met by an e-learning approach alone. Recognising this as one of the major shortcomings of the ENeA SEA programme as it has been developed up to date, partners seek to further implement a strong complementing face-to-face case based training part of the programme. The developed tutor area and its materials described earlier will serve as a starting point for these ambitions. A currently planned roll-out of ENeA SEA in Indonesia will additionally develop and execute a concept for evaluating the actual public health impact mediated by such an educational programme for HCPs on the ultimate target groups ((pre-)pregnant women, infants, young children and their families).

## 5. Conclusions

We conclude that the ENeA SEA e-learning programme contributes successfully to improving practical counselling competencies targeting (pre-) pregnant women and families of infants and young children. This meets national and regional ambitions to reduce early nutrition and lifestyle-related health problems and subsequent risks of suboptimal development and physical and cognitive abilities, and of NCD risk. The launch of the ENeA SEA platform customised curriculum also allows targeted e-learning for effective and efficient knowledge acquisition within limited time availabilities. Exploitation and sustainability actions with key stakeholders should ensure updates of e-modules and their integration into pre-and postgraduate study programmes. Inclusion in CME/CPD systems in Thailand and Malaysia will be fostered by local language versions of the modules. We aim at further expansion into other SEA countries, such as Indonesia. We anticipate that these actions will support reducing the double burden of malnutrition and obesity that arises through suboptimal early nutrition and are a major public health concern.

## Figures and Tables

**Figure 1 nutrients-12-01817-f001:**
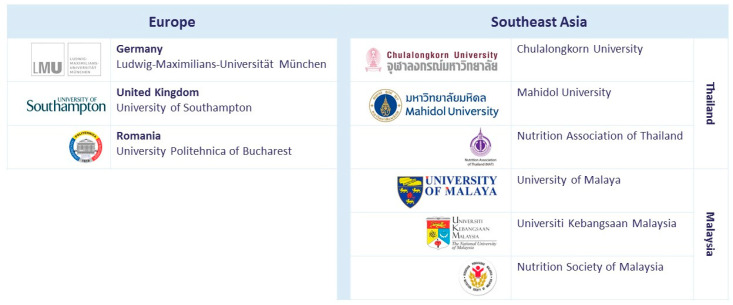
Early Nutrition eAcademy Southeast Asia (ENeA SEA) consortium partners.

**Figure 2 nutrients-12-01817-f002:**
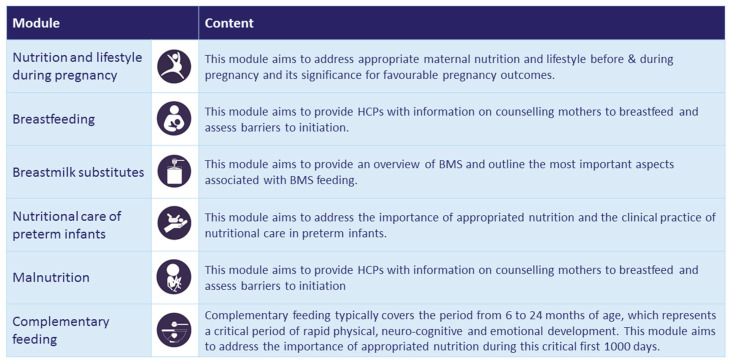
ENeA SEA e-learning modules and their content.

**Table 1 nutrients-12-01817-t001:** Questions and participant responses: closed online questionnaire to health care professionals (HCPs).

Question	Responses
Q1: Please rank the following topics on a scale of 1–5 (1 being most important and 5 being least important; each ranking number from 1–5 can only be used once) in which doctors in practice and other health care professionals (HCP) require continual medical education?	The topics “nutrition and lifestyle during pregnancy” (*n* = 61/142) and “breastfeeding” (*n* = 43/142), and nutritional care of preterm infants (*n* = 23/142) were reported as the most required topic areas (rated with 1) for HCPs to receive continuing medical education (CME).
Q2: In your opinion, do doctors in practice and other HCPs use the internet for professional purposes?”“In your opinion, are doctors in practice and other HCPs proficient enough in English to study an online course?”	85–91% of HCPs reported using the internet for professional purposes. Doctors more often stated to be proficient in English to study online (89%) than other HCPs (63%).
Q3: How are the doctors in practice and other HCPs continually trained and educated in their area of study?	Most of the existing training for doctors (83%) and HCPs (85%) was at face-to-face events; e-learning was reported by slightly more doctors (41%) than other HCPs (35%).
Q4: Out of the following, which formats are preferred for training for doctors in practice and other HCPs?	Face to face events were the preferred mode of training for both doctors (72%) and other HCPs (76%). 57% of doctors and other HCPs preferred e-courses for their training.
Q5: Out of the following, which modes are preferred for e-learning for doctors in practice and other HCPs?	Availability on a smartphone, laptop or mobile phone were the preferred modes of e-learning, with fewer preferring a computer.
Q6: Are doctors in practice and other HCPs evaluated on a continuing medical education/professional development (CME/CPD) credit system in your country?	81–85% of both doctors and other HCPs reported that they were evaluated on a CME/CPD system.
Q7: On a scale of 1–5, how important are CME/CPD credits for you (1 being highly important and 5 being least important)?	37% of the doctors and other HCPs rated the importance of CME/CPD credits as very high; 21% rated the importance as high.
Q8: On what basis are doctors in practice and other HCPs evaluated?”	A credit point-based system was slightly more relevant to HCPs (73%) than doctors (64%).
Q9: What types of CME/CPD certificates are recognised in your country?	97% of doctors and 96% of other HCPs reported that national/local certificates were recognised. 11% of doctors and 7% of other HCPs stated that European (UEMS) and US American (AMA) CME credits were recognised in their countries.

**Table 2 nutrients-12-01817-t002:** Questions and participant responses in the open questionnaire to stakeholders with an influencing role on CME/CPD regulations.

Question	Responses
Q1: What knowledge, skills and competencies should the specialised doctors/health care professionals have in your country?	54% of the responses related to nutrition and lifestyle, with 90% stating that it could be further clustered into the themes pregnancy, infant formula feeding, breastfeeding and complementary feeding.
Q2: What knowledge, skills and competencies are currently lacking in the specialised doctors/health care professionals according to your opinion?	85% of the responses related to nutrition and lifestyle, with 90% stating it that could be further clustered into the themes pregnancy, preterm infants, infant formula feeding, and breastfeeding.
Q3: Training in early nutrition content: what contents should be included in the specialisation and CME/CPD programs?	20% of the topics identified related to complementary feeding, 19% to infant formula feeding, 18% to breastfeeding, 15% to preterm infant nutrition and 11% to critical nutrients.
Q4: Is nutrition related content included in specialisation training for doctors/health care professionals? If yes, what percentage is dedicated to nutrition?	10 out of 11 participants responded affirmatively. Nutrition content ranged from 5–10%.

**Table 3 nutrients-12-01817-t003:** User statistics per module and unit of ENeA SEA e-learning (as of 24 April, 2020).

Module	Nutrition and Lifestyle in Pregnancy					Breastfeeding			Breastmilk Substitutes				Complementary Feeding				Nutritional Care of Preterm Infants		
Total of enrolled users	421					636			266				293				216		
Users attempted CME test (*n*)	46					378			140				184				116		
Users successfully passing CME test (*n*)	38					262			95				150				93		
Passing rate ^1^ (%)	83%					69%			68%				82%				80%		
Unit	1	2	3	4	5	1	2	3	1	2	3	4	1	2	3	4	1	2	3
Enrolled users in each unit (*n*)	306	160	132	103	91	577	430	412	210	150	157	153	248	223	209	241	185	154	152

^1^ Passing rate = number of users successfully completing the CME test/number of users with at least 1 CME test attempt.

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
