# Peer review of "Early Nutrition eAcademy Southeast Asia e-Learning for Enhancing Knowledge on Nutrition during the First 1000 Days of Life"

_nutrients, 2020, doi:10.3390/nu12061817_

Round 1

Reviewer 1 Report

Early Nutrition eAcademy Southeast Asia e-learning 3 for enhancing knowledge on nutrition during the 4 first 1,000 days is an interesting study however the authors should include first 1000 days of what in the title. This study provides another E learning programs focussing on nutrition and lifestyle during the first 1,000 days of life especially for the SEA region, which has its advantages and disadvantages. While the advantages are well documented in the manuscript, I would recommend the authors to include the shortcomings and suggest alternatives of this program in the manuscript as well. Otherwise, the manuscript is a well thought out study in my opinion.

Author Response

Dear Reviewer,

thank you very much indeed for those valuable comments, please find the answers to each comment as replies directly afterwards:

Reviewer comment 1:

Early Nutrition eAcademy Southeast Asia e-learning 3 for enhancing knowledge on nutrition during the 4 first 1,000 days is an interesting study however the authors should include first 1000 days of what in the title.

Reply by the author: thank you very much for alerting this, the title has been revised.

Reviewer comment 2:

This study provides another E learning programs focussing on nutrition and lifestyle during the first 1,000 days of life especially for the SEA region, which has its advantages and disadvantages. While the advantages are well documented in the manuscript, I would recommend the authors to include the shortcomings and suggest alternatives of this program in the manuscript as well.

Reply by the author: shortcomings have been added and explained and alternatives for further optimization suggested.

Thank you very much and best regards,

Brigitte Brands

Reviewer 2 Report

The article presents a work of extreme relevance for the development of skills of health care professionals and for the promotion of health during the first 1,000 days. The study methodology is well described as well as the results of the development of the e-learning project.

There are minor issues that would be important to address in order to improve the manuscript.

Lines 81 - 92: I believe that this text does not belong to the manuscript.

l. 160: Bearing in mind that health professionals were recruited through personal and professional contacts, was there any verification of the registrants? Have inclusion criteria been established?

l. 218 Adding the numbers presented gives 124 and not 142 professionals. What is the justification? If there are other professionals from different professional categories, they should be included in a separate category, perhaps called "other professionals".

l. 252 - should be CME instead of SME

l. 318: Is there any verification of the professional or his professional category (perhaps a professional number?) Is the training open to everyone?

Is there any knowledge assessment? Is the training assessment based solely on the qualitative assessment of the participants? It must be explained.

l. 326- 327: What is the justification that the authors find for "the number of enrolled users in each module is not as high"?

Author Response

Dear reviewer,

thank you very much for those contructive comments. Please find the answers to each comment directly underneith in the section below. All changes made to the manuscript according to our replies are highlighted in yellow in the re-submitted document.

Reviewer comment 1: Lines 81 - 92: I believe that this text does not belong to the manuscript.

Reply by the author: Thank you very much, yes, this has been accidentially not deleted from the manuscript template. This has now been deleted.

Reviewer comment 2: l. 160: Bearing in mind that health professionals were recruited through personal and professional contacts, was there any verification of the registrants? Have inclusion criteria been established?

Reply by the author: Inclusion criteria for being eligible to fill-in the online questionnaires were set by the mandatory selection of a health care profession from a drop down menu. In addition, only HCPs from the SEA region were included in the analysis. There were no further inclusion restrictions to allow for a higher number of participation and to retrieve a broader spectrum of answers.A respective sentence has been added to the manuscript now.

Reviewer comment 3: l. 218 Adding the numbers presented gives 124 and not 142 professionals. What is the justification? If there are other professionals from different professional categories, they should be included in a separate category, perhaps called "other professionals".

Reply by the author: Thank you very much for detecting this missing information. The last category “other health care professionals” has been added to complete the sample description in the manuscript.

Reviewer comment 4: l. 252 - should be CME instead of SME

Reply by the author: This has been corrected now.

Reviewer comment 5: l. 318: Is there any verification of the professional or his professional category (perhaps a professional number?) Is the training open to everyone?

Reply by the author: While in principal and technically open to everyone, the ENeA SEA platform and its modules are designed and to be used by health care professionals only. A corresponding tick-box has to be confirmed actively by the user while registering to the platform. This is also contained in the privacy statement and use of terms of the platform.

Reviewer comment 6: Is there any knowledge assessment? Is the training assessment based solely on the qualitative assessment of the participants? It must be explained.

Reply by the author: While the collection of feedback from evaluation questionnaires is a valuable information source for further improving the ENeA SEA modules and programme, knowledge acquisition by the user is evaluated by the embedded CME tests with multiple choice questions. Moreover, passing the CME tests with a threshold of 70% of correct answers is a pre-requisite to obtain the CME Certificates of Completion. This passage has now been added to the manuscript.

Reviewer comment 7: l. 326- 327: What is the justification that the authors find for "the number of enrolled users in each module is not as high"?

Reply by the author: This could be explained by the “on-demand” learning nature of the platform which allows to start a module at any time after having registered on the platform. This explanation has now been added to the manuscript.

Thank you very much and best regards

Brigitte Brands